# Nanosafety: An Evolving Concept to Bring the Safest Possible Nanomaterials to Society and Environment

**DOI:** 10.3390/nano12111810

**Published:** 2022-05-25

**Authors:** Filipa Lebre, Nivedita Chatterjee, Samantha Costa, Eli Fernández-de-Gortari, Carla Lopes, João Meneses, Luís Ortiz, Ana R. Ribeiro, Vânia Vilas-Boas, Ernesto Alfaro-Moreno

**Affiliations:** 1NanoSafety Group, International Iberian Nanotechnology Laboratory, 4715-330 Braga, Portugal; filipa.lebre@inl.int (F.L.); nivedita.chatterjee@inl.int (N.C.); eli.fernandez@inl.int (E.F.-d.-G.); carla.lopes@inl.int (C.L.); joao.meneses@inl.int (J.M.); ana.ribeiro@inl.int (A.R.R.); vania.vilasboas@inl.int (V.V.-B.); 2Masters in Biophysics and Bionanosystems, Campus de Gualtar, School of Sciences of the University of Minho, 4710-057 Braga, Portugal; samanthacosta1996@hotmail.com; 3Research Centre for Experimental Marine Biology and Biotechnology (PIE-UPV/EHU), 48620 Plentzia, Spain; luismauricio.ortizgz@outlook.com

**Keywords:** nanomaterials, nanotoxicology, immunotoxicity, genotoxicity, epigenetics, advanced in vitro models, in silico, life cycle assessment

## Abstract

The use of nanomaterials has been increasing in recent times, and they are widely used in industries such as cosmetics, drugs, food, water treatment, and agriculture. The rapid development of new nanomaterials demands a set of approaches to evaluate the potential toxicity and risks related to them. In this regard, nanosafety has been using and adapting already existing methods (toxicological approach), but the unique characteristics of nanomaterials demand new approaches (nanotoxicology) to fully understand the potential toxicity, immunotoxicity, and (epi)genotoxicity. In addition, new technologies, such as organs-on-chips and sophisticated sensors, are under development and/or adaptation. All the information generated is used to develop new in silico approaches trying to predict the potential effects of newly developed materials. The overall evaluation of nanomaterials from their production to their final disposal chain is completed using the life cycle assessment (LCA), which is becoming an important element of nanosafety considering sustainability and environmental impact. In this review, we give an overview of all these elements of nanosafety.

## 1. Introduction

The concepts of nanomaterials and nanotechnology were developed in the 20th century, and from the birth of quantum theory, it was evident that the physicochemical properties of matter in the nanoscale (i.e., nanomaterials) would be challenging to handle and understand [1,2]. Although examples of nanomaterials can be found in ancient products [3], it was not until recently that the knowledge and understanding of these materials came to light. Nanotechnology was first suggested by Richard Feynman in the late 1950s [4], and in 1974, Norio Taniguchi coined the term “nanotechnology” [5]. With the discovery of fullerenes in 1985, the revolution of nanomaterials gained momentum, and presently we talk about carbon nanotubes (CNTs), graphene, quantum dots, nanorods, nanowires, nanofibers, and all sorts of composites that include different types and other materials [6].

Nanosized structures have been already in commercial use for several years. Cosmetics, drug delivery, food, water treatment, and agriculture are among the industries that use a large variety of these types of structures, including liposomes, magnetic nanoparticles, and titanium dioxide (TiO_2_) [7]. We also must consider that the accidental and intentional release of nanomaterials into the environment is a growing concern [8]. For these reasons, exposure to nanomaterials can be related to multiple routes, and the differences in type (chemical composition, physical properties, shape, and size) and sources give a very complex dimension to the task of assessing their safety and potential risks [9]. Just as an example, recently it has been reported that plastics of submicron size were found in the bloodstream of healthy donors [10]. This finding is not surprising considering that the translocation of particles in that size range was reported in experimental studies in 2002 [11], and since then, many other studies have shown that particulate matter can translocate to the circulation [12] and can reach different tissues, including the brain [13], even in a fetal stage [14]. The difference between this study of plastics in blood and previous reports is that the plastics report comes at a time when the discussion related to microplastics in the oceans [15] has reached the mass media [16], therefore giving great mediatic attention to this report [17].

The production and use of nanomaterials can be found in a wide variety of products, from components for electronic devices to cosmetics and food [18]. The potential impact on the environment is also a problem to consider, due to the fact that the final disposal can lead to contamination of soil, water, and air [19]. However, what are the routes of exposure to nanomaterials? In the following image, we try to compile the main and most common routes of exposure (Figure 1).

The impact of nanotechnology and nanomaterials is transforming the world, and the following numbers can give us an overview of how nanotechnology and nanomaterials are becoming central players in our economies. In recent years, the development of nanotechnology and the use of nanomaterials have experienced exponential growth, and it is estimated that in 2022 the market of these nanomaterials may reach a revenue of USD 9.1 billion in Europe only [20], and other estimations calculate that by 2024 the worldwide market of nanomaterials may reach USD 125 billion [21]. In 2018, there were more than 11,000 new applications for patents related to nanotechnology in the United States and more than 1700 in the European Patent Office [22].

Since ancient times, it has been known that inhalation, ingestion, and skin exposure to certain substances can be deleterious to human health, and miners are probably the best example in history of occupational exposure to materials leading to health problems. This has been reported since the time of the Roman Empire [23]. It is said that “those who cannot remember the past are condemned to repeat it”, and, in this regard, there has been growing concern about how nanomaterials could be deleterious for workers that are exposed to these materials [24] and hence the need to address their safety. There is also a growing concern related to how the final users of these products may be exposed and potentially experience adverse outcomes. In order to tackle these issues, different strategies have been implemented to evaluate the toxicity of nanomaterials, which is the first step in assessing the potential risk that workers, final users, and the environment may encounter. Environmental [25], animal [26], in vitro [27], and in silico [28] models are used to assess the potential effects of nanomaterials, and many of these models are adaptations of previous strategies used to evaluate chemicals or larger particles. This type of adaptation has created problems with accuracy and confidence in the obtained results, considering that nanomaterials may interfere with the traditional methods.

In this review, our aim is to give an overview of the state of the art of the main approaches related to the evaluation of nanomaterials in relation to toxicity and safety and the emerging strategies related to the field. For this, we present the advances in nanotoxicology, immunotoxicology, epigenetics and genotoxicity, advanced models such as organs-on-chips, cheminformatics/in silico tools, and LCA.

## 2. Nanotoxicology

The assessment of the detrimental effects of nanomaterials on living organisms, known as nanotoxicology, is still mostly an extension of the conventional methods used to assess the toxicity of other chemicals (including drugs) in vivo and in vitro. An accurate assessment of the toxicological effects of nanomaterials depends on having good knowledge of their characteristics [29], namely composition, size, shape, dispersity, surface charge, surface functionality, protein corona formation, etc. These are well-known determinants of a material’s ability to cross biological barriers, as well as of its agglomeration/aggregation status [30], which, in turn, brings additional challenges. These manifest, for example, as interference with the optical readouts or direct chemical reaction with assay components of conventional in vitro techniques [31]. Table 1 summarizes reported interferences of nanomaterials with conventional in vitro assays and suggests possible solutions for the identified interference.

Besides the methodological issues identified above, the reported exposure/dose metrics are another major concern that has been the subject of debate [32]. This is because a mass-based dose metric (e.g., µg·cm^−3^), a concept introduced in the context of in vivo exposure to airborne particulate matter, is still conventionally used in nanotoxicology. However, this metric experiences substantial impact from small fluctuations in dispersity values of the nanomaterials, confirming its inappropriateness for the purpose. Reporting the number of nanoparticles or the particles’ surface area is seemingly more representative in some cases [29,33,34,35,36]. In fact, these metrics were found to correlate well with increased mortality [34] and with the occurrence of adverse outcomes such as inflammation, both in vivo [36] and in vitro [33,35]. More recently, a combination of two criteria, namely particle size together with mass or surface area, was identified as more predictive of the occurrence of lung toxicity derived from nanoparticles, as it allows overcoming the limitations of individual dose metrics [37].

Still, differences in the size, density, and surface reactivity of nanomaterials imply that they dissolve, settle, diffuse, or agglomerate differently as well [38]. Therefore, while the metrics referred to above describe the exposure conditions well, they do not adequately reflect the dose of nanomaterials that effectively interacts with tissues or cells. For that, the contact surface area (of the cells in culture or of the exposed tissue in vivo) needs to be accounted for to reflect the delivered dose per surface area or mass [39,40]. Ultimately, this will allow the estimation of the cellular dose with the help of highly sensitive analytical methods and computational tools [38,41,42]. Noteworthily, all the above-cited works relate to pulmonary toxicity either in vivo or in vitro, as the lung is the subject of most of the studies on both particulate matter and nanomaterials (Figure 2). Although inhalation is considered the main route of exposure to nanomaterials, these materials can be translocated to the systemic circulation [11], reaching other organs where they end up accumulating [43,44,45]. Therefore, it would be important to verify whether similar correlations between the surface area metric and the toxicological outcomes are also observed for other important target organs, such as the liver and the brain, and other exposure settings, types of nanomaterials, and outcomes of interest. The conflicting results obtained from different laboratories additionally highlight an urgent need for the development of standard protocols for handling nanomaterials and testing them in biological systems detailed to the level of how to apply nanoparticles to the cell cultures [46]. This would be a crucial step in improving the current status of nanotoxicological assessments.

The levels of exposure to nanomaterials in occupational, accidental, or other scenarios are yet another fundamental aspect of adequately identifying proper dose ranges to be tested for nanotoxicology purposes. For that, a multidisciplinary approach is required in order to develop tools that allow for such characterization. In line with this, the Horizon Europe program has already granted funding to develop low-cost sensors to monitor the levels of ultrafine particles in closed environments [47]. Despite the great step forward, this will still be a conservative approach, and an estimated internal exposure dose would be ultimately desired [48], as not the whole exposure dose will effectively interact with the target tissues. For now, most published studies mainly address acute exposures to unrealistically high doses of nanomaterials. While these data are extremely important for understanding the hazard and the mechanisms by which nanomaterials may exert toxicity, the lack of translatability to real-life exposure scenarios fails to convey the real risk posed by nanomaterials. Important first steps are being already taken as attempts to overcome this issue by undertaking longer-term, repeated exposures to lower than usually tested concentrations of nanomaterials both in vivo [49,50] and in vitro [51,52,53]. Moreover, the current investment in the development of new, human-based, more advanced in vitro models with prolonged cultivation times (see Section 5) will allow for repeated exposures to low concentrations, which would more closely resemble the most common exposure scenarios. Still, we are far from testing realistic conditions, where we are continuously and simultaneously exposed not only to nanomaterials but to many different chemicals that may synergize, potentiate, or inhibit each other [54]. Thinking forward, predicting internal exposure doses to multichemical mixtures and their effects would be a tremendous achievement for (nano)toxicology. Because it is realistically impossible to experimentally test every nanomaterial type, size, and shape, it becomes fundamental that experimentalists and computational scientists closely collaborate in the search for common terminology, paving the way toward a more predictive nanotoxicology.

A major ambition of nanotoxicology is the understanding of the mechanistic basis behind the adverse effects induced by nanomaterials. From the knowledge collected so far, some mechanisms stand out as the most frequently uncovered for different types of nanomaterials, namely oxidative stress, inflammation, and deoxyribonucleic acid (DNA) damage leading to cell death [44]. Even though the sequence of events is not yet clear or fully understood, it is likely that the overproduction of reactive oxygen species (ROS) is, in fact, the trigger leading to a sequence of events that culminates in cell death (Figure 3).

The described mechanisms have been associated with nanomaterial-induced adverse effects in the liver [55] and the brain [56], among other organs, with important repercussions observed in the offspring as well [57]. Compared to bigger-sized homologous materials, the enlarged surface area of nanomaterials comes with increased reactivity, which seems to be the main driver for the observed overproduction of ROS. The impact of nanomaterials’ characteristics on their cytotoxicity is the subject of extensive in silico studies and is further addressed in Section 6.

Together with a robust characterization of the nanomaterials [58], other recognized primary needs for the advancement of nanotoxicology are higher-throughput assays and the development of standards and test guidelines to fulfill regulatory demands [59]. Certainly, it is also fundamental to properly communicate the findings to the nonscientific community to avoid a general perception of risk that is yet to be fully substantiated, preventing bias towards fear of and a still unwarranted fight against the use of nanomaterials.

## 3. Immunotoxicity

As mentioned earlier, the inflammatory response, driven by the innate immune system, plays a major role in nanomaterial-induced toxicity. During the last couple of decades, a wealth of compelling evidence has been gathered to support the idea that immune responses to biomaterials have a great impact on their biological activity and toxicological effects [60,61,62,63]. Industries are designing more advanced materials, which means that more workers and consumers will be exposed to nanoscale particles, posing a risk of detrimental effects on natural systems and ultimately on humans. For instance, it is well documented that exposure to fine inhalable pollution particles is directly linked to millions of deaths [64,65], and Shears et al. showed that exposure to diesel exhaust particles increases susceptibility to invasive pneumococcal disease [66]. However, there are large knowledge gaps regarding the potential long-term health effects associated with these particles. It is imperative to reassess the impact of nanoparticulate material on short- and longstanding detrimental health effects, and thus more in-depth knowledge of nanoparticle interaction with the immune system is required (Figure 4). Immunotoxicology is a recent subfield of toxicology that studies the deleterious immunomodulatory effects (e.g., immunosuppression, immunostimulation, hyperinflammation) of newly developed and existent (nano)materials. The growing relevance of this field is patent in the recently published technical report ISO/TR 10993–22:2017 which has a subsection devoted to immunotoxicity, on the toxicological evaluation of medical devices that are composed of or contain nanomaterials [67].

While studying the immunomodulatory effect of a nanomaterial is of the uttermost importance to be aware of possible endotoxin contamination, a fundamental barrier that can cause serious toxic effects in humans [68,69] and may give rise to confounding or even opposite results in the literature. To give an example, one group showed that chitosan had the ability to inhibit lipopolysaccharide (LPS)-induced secretion of proinflammatory cytokines [70], while a separate group observed the reversed effect [71]. However, when using endotoxin-free chitosan, the polymer on its own was incapable of generating proinflammatory cytokine secretion [72]. The release of LPS, present on the outer membrane of Gram-negative bacteria [73], takes place after death and lysis of the cell; since microorganisms are ubiquitously present during nanomaterial preparation, unless precautionary measures are taken into consideration, the risk of inadvertent contamination is high. At the present, the Limulus Amebocyte Lysate (LAL) assay is considered the golden standard for assessing and quantifying endotoxin content in pharmaceutical products and medical devices; nevertheless, there are a few limitations for its application [74]. For that reason, it is important to use a second method to validate our results, such as the monocyte activation test [75], especially in the case of preparations containing nanomaterials, considering that interferences with toxicological assays are a concern [76], as discussed in the previous section.

It is imperative to design more sophisticated in vitro models that can better recapitulate complex aspects of human physiology. This is of particular importance in the cancer field, where it was shown that a staggering 97% of drugs that are in clinical trials fail, as they may not work on the targets researchers intended [77]. One reason that helps to explain these results, and that will be addressed in more detail in Section 5, is the use of simplified static two-dimensional (2D) monoculture models (e.g., NCI-60 cell line screening panel [78]), which fail to recapitulate physiologically relevant in vivo mechanisms, thus worsening in vitro translation to clinical data. In this context, the inclusion of cells of the immune system (e.g., macrophages, monocytes, lymphoid cells) during preclinical studies has the huge potential to facilitate the identification of clinical candidates with cytotoxic capacities, minimize iterative steps or large and prolonged trials, and increase the probability of regulatory success, since many immune cells are involved in events that support tumorigenesis [79]. This is highly appealing for the pharmaceutical industry since the low successful translation of drug candidates in research into approved products has a major economic impact [80,81]. Thus, the inclusion of immune cells in more preclinical models should become a routine methodology. Of note, the selection of cell lines as models of cells of the immune system is still challenging. For instance, macrophages have a pivotal role in the host response to foreign materials; the two most used cell lines for macrophage studies are THP-1 and U937 cells. While they are generally good models to study gene expression and cytokine secretion upon exposure to a myriad of materials, they fail to recapitulate important aspects present in blood-derived macrophages [82,83].

Recently, a connection between nanomaterials and immune training has emerged. Vertebrate immunity is classically divided into innate and adaptive immune responses. The innate system is typically characterized as a nonspecific and rapid first line of defense, yet recent findings have challenged this classical view, supporting the idea that the innate immune system is able to be programmed to generate an enhanced nonspecific response upon a later challenge, in a process termed “trained immunity”. It has been demonstrated that this protection is achieved through epigenetic and metabolic reprogramming which allows for stable, long-term chemical alterations in the DNA that modify the transcriptional potential of a cell by regulating gene expression [84,85]. This idea that innate cells can retain some memory of past immunological insults, allowing for enhanced cellular responses to secondary infections, is well documented in the case of microbial stimuli [86,87] and endogenous molecules [88,89] but has only recently been demonstrated for exogenous molecules [90,91]. Little information is available about the capacity of nanomaterials to promote this phenomenon, making it an exciting field, with immense possibilities. It is fair to hypothesize that more nanomaterials are able to induce innate immune cell reprogramming. While trained immunity can prime the body to generate more effective immune responses, it can also have deleterious consequences contributing to hyperinflammatory states or inadequate responses to future challenges, thus modifying the capacity of an organism to adapt to the environment. Nanomaterials’ innate training capacity has been overly neglected. Systematically assessing the immune profile generated by distinct nanomaterial attributes and correlating those to the overall impact on the immune system might endow us with the capacity to predict the immunosafety of a novel and existing materials more accurately and lay the basis for generating three-dimensional (3D) models incorporating multiple cell types, including immune cells, that better recapitulate in vivo complexity.

Addressing the immunomodulatory abilities of nanomaterials adds scientific and economical value, shedding light on potential harmful consequences of existing substances, guiding the design of novel and safer materials, and providing fertile ground for the development of new therapeutic agents.

## 4. Genotoxicity and Epigenetics

### 4.1. Nanogenotoxicity

Genotoxicity, the damage to genetic materials, may lead to carcinogenesis and other chronic diseases. If germ cell DNA is compromised, it will affect the individual health and could also have an impact on the next generations [92,93,94]. Therefore, genotoxicity assessment is considered a crucial aspect of nanomaterial hazard identification. The mode of nanomaterial-induced genotoxicity can be classified as direct primary (interaction between nanomaterial and genetic materials directly), indirect primary (nanomaterial-induced reactive nitrogen species (RNS)/ROS species affecting the genetic materials), and secondary (damage of genetic materials due to nanomaterial-induced inflammation) mechanisms [92,95,96,97]. Oxidative stress has been widely considered an underlying mechanism of nanomaterial-induced genotoxicity [92,94,98,99].

The current state of genotoxicity assessments still depends on genotoxicity tests that were designed to screen general chemical agents and have been adapted for nanomaterials (Table 2), and no nanoparticle-specific positive controls have been established [100] in this regard. The genotoxicity assessment mainly includes DNA damage (strand break, adduct formation), gene mutation, and chromosomal damage (clastogenicity and aneugenicity) as endpoints [92,93,94,98]. In vitro (human and murine cultured cell lines) systems are the most widely used models, followed by in vivo (mainly rats and mice) ones, for nanogenotoxicity assessment [93,98]. In vitro models are appropriate for primary genotoxicity assessment, while in vivo systems are more suitable for secondary genotoxicity. The ongoing efforts on advanced in vitro system (e.g., 3D coculture of different cell types) development have the potential to simulate the in vivo microenvironment and successfully assess secondary genotoxicity [94,95]. In addition, ecotoxicological models, namely fish [101,102], drosophila [103], nematodes [104], yeast, bacteria [105], etc., are also in use for nanogenotoxicity assessment (Figure 5). The largely applied assays for nanogenotoxicity assessment are comet assays followed by micronucleus (MN) assays. The confirmation and standardization of nanomaterial-specific genotoxicity assessments are still ongoing processes; however, in the interim, in vitro based comet assays, MN assays, and hypoxanthine-guanine phosphoribosyltransferase (HPRT) assays are recommended as useful standard battery test methods for nanogenotoxicity assessment with the confirmation of uptake of nanomaterials [93,98,99]. Furthermore, a modified comet with DNA repair enzymes (8-oxoguanine DNA glycosylase (OGG1) and formamidopyrimidine-DNA glycosylase (FPG)) has been used to address oxidative stress-mediated DNA damage [98]. The nanogenotoxicity field and related test systems are continuously evolving, improving [93], and adapting from basic biology. Examples include (i) the detection of alterations in expression of sensors (genes/proteins) for DNA damage response and checkpoint pathways (namely phosphorylation of the Ser-139 residue of the histone variant (γH2AX), protein p53 binding protein 1 (53BP1), ataxia–telangiectasia mutated serine/threonine kinase (ATM), ataxia–telangiectasia and Rad3-related serine/threonine kinase (ATR), tumor protein p53 (p53), cyclin-dependent kinase inhibitor 1A (p21), checkpoint kinase 1 (CHK1), CHK2) and (ii) the establishment and application of reporter cell lines related to DNA damage response and DNA repair pathways, such as the ToxTracker reporter assay [106,107]. An obvious future potentiality is not only applying new genotoxicity assays but also upgrading existing assays to high-throughput (HT) and high-content (HC) platforms, such as HT comet assays, HT in vitro MN assays, and HC γH2AX or 53BP1 assays [106,108]. In addition, the multiplex detection (e.g., Luminex-based platform) of biomarkers from DNA damage response and repair pathways could potentially improve future nanogenotoxicity assessments. A great diversity in testing strategies, model systems, and results is evident in nanotoxicity assessments. Some studies also highlighted that the selection of cell lines could affect the results of the same nanomaterials [98].

It is noteworthy that most focus has been given to DNA damage (strand break analyzed by comet assay) or oxidative stress mode action (8-hydroxydeoxyguanosine (8-OHdG) detection or modified comet assay with OGG1 and FPG enzymes). The other mode of DNA damage mechanisms, including perturbation of DNA repair or synthesis processes, has been widely neglected in the nanogenotoxicity field [98,112,113]. Altered DNA repair (enhanced or reduced) has been linked to various diseases, including cancer [112]. Among the few studies focused on the adverse effects of engineered nanomaterials in the DNA repair process are studies showing that silver nanoparticles (AgNPs) induce nuclear factor erythroid-related factor 2 (Nrf-2)-mediated downregulation of OGG1 gene expression [114] and affect the nonhomologous end-joining (NHEJ) repair pathway through targeting DNA-dependent protein kinase, catalytic subunit (DNA-PKcs) [115], in human cell lines. In addition, perturbed base excision repair (BER) and nucleotide excision repair (NER) abilities in human cell lines exposed to TiO_2_ and AgNP were also reported recently [109,110]. In the same line of evidence, AgNP exposure affects the BER repair pathway (OGG1, nudix hydrolase 1 (MTH1)) in human cell lines and affects Bis(5′-nucleosyl)-tetraphosphatase (NDX-4) in *Caenorhabditis elegans* as a function of p38 mitogen-activated protein kinase (p38MAPK) [104]. Recently, a fluorescence multiplex−host-cell reactivation (FM-HCR) assay system has been applied to evaluate the nanomaterial-induced altered DNA repair capacity and can assess all six major DNA repair pathways in a single assay platform [112,116]. Assay development and refinement are vital for understanding the underlying mechanism of genotoxicity. The nanogenotoxicity assessment can be improved by integrating the methodological approaches based on the evaluation of DNA repair capacity and DNA strand break as well as upgrading to HT platforms.

Understanding the test nanomaterials and their physicochemical properties and interactive behavior in exposed systems can be critical in determining the genotoxic potentiality [96,99]. Hence, the selection of the appropriate assay can be based on the nanomaterial’s intrinsic characteristics and exposure-system-dependent properties [98].

### 4.2. Nanoepigenetics

Epigenetics is the field of study which investigates the modification of gene expression without changes in DNA sequences. The term “epigenetics” has been widened by the National Institutes of Health (NIH) as “both heritable changes in gene activity and expression but also stable, long-term alterations in the transcriptional potential of a cell that are not necessarily heritable” [117]. As the changes can be stable, reversible, or heritable, the exposure/effects of early life can influence later life disease susceptibility or even pass through the successive unexposed generations. The primary epigenetic marks involved in gene regulations are DNA methylation, histone modifications, and noncoding ribonucleic acid (ncRNA) (e.g., long noncoding RNA (lncRNA), microRNA (miRNA), small interfering RNA (siRNA), piwi-interacting RNA (piRNA), circular RNA (circRNA)) and chromatin remodeling. Recent studies indicate that the epigenome and epigenetic regulatory mechanisms play a pivotal role in gene–environment interactions shaping the phenotype, including adaptive response and adverse outcomes, along with various diseases [118,119,120]. The alterations of epigenetic biomarkers are highly dynamic and depend on various factors such as species, exposure, time, and even cell/tissue type. Epigenetic biomarkers’ dynamic nature and plasticity make the development of testing strategies highly challenging. At present, there is no single platform available to screen all the known epigenetic modifications to identify the most important or altered epigenetic marker linked to exposure to a particular chemical. Alterations in targeted epigenetic markers can be measured with existing standard molecular biology techniques and whole-genome sequencing (Table 3).

Accumulating evidence supports incorporating epigenetic biomarkers as toxicity endpoints, including in nanomaterial risk assessment. Nonetheless, the current knowledge is insufficient with respect to the role of each piece of epigenetic machinery in gene expression in normal and complex diseased conditions. In other words, there is still a lack of clarity on the role of epigenetic biomarkers in the causality of adverse health outcomes [135]. Therefore, a better understanding of the adverse effects of nanomaterials on the epigenome is necessary. Currently, most nanomaterial-induced epigenetic toxicity studies focus on DNA methylation changes and the related enzyme machinery. DNA methylation mainly includes 5-methylcytosine (5mC), 5-hydroxymethyl cytosine (5hmC), and 6-adenine methylation (6mA) markers (Figure 6). Most nanomaterial-induced DNA methylation alterations were reported to be related to the 5mC and 5hmC markers; specifically, two studies demonstrated altered 6mA levels in rats exposed to carbon black nanoparticles [136,137]. The other most studied epigenetic biomarkers are miRNA profiling changes due to nanomaterial exposure (Figure 6). In particular, some studies reported integrative analysis of ncRNA with transcriptomes (mRNA) [132,138] or proteomes [139].

Conversely, only a handful of studies focused on histone modification or lncRNA expression profiling in nanomaterial-exposed conditions. The study of circRNA, another type of ncRNA, is just gaining momentum in the toxicity field and in nanomaterial hazard identification assessment [133,140]. The increasing number of studies demonstrating the epigenetic alterations resulting from nanomaterial exposure still lacks a connection with potential health risks. This is due to the unavailability of epigenetic modifications in epidemiological studies, except for only a few related to workplace nanomaterial exposure [123,132,141,142,143]. Like nanogenotoxicity, the assessment of the epigenetic toxicity of nanomaterials is mostly based on in vitro followed by in vivo (mice or rats) models (Figure 6). Besides these models, DNA methylation studies were also reported in ecotoxicology model species (such as white worms [144] and zebrafish [145,146]), while histone modification [147] and ncRNA (lncRNA and miRNA) were mainly reported in nanomaterial-exposed roundworms (*C. elegans*) (Figure 6). In the epigenetic study design, one must consider the development of a long-term impact with multigenerational or transgenerational effects, preferably with alternative in vivo models (e.g., zebrafish, drosophila, and *C. elegans*) [147] to reduce animal use. Furthermore, epigenetic alterations in adverse effects vs. adaptive response in nanomaterial-treated conditions also need to be considered; however, this can only possibly be achieved with future technological advancement and improved testing strategies.

In summary, integrating epigenetic endpoints in nanomaterial risk and safety assessment, with all of the current limitations and existing challenges, needs considerable attention. However, there is a need for a sufficient amount of reproducible data on the causal relationships between nanomaterial exposure, adverse phenotypes, and specific epigenetic markers [94,148]. Therefore, we have to travel a long way to select the most representative epigenetic biomarkers to be evaluated with the most reliable testing strategies in the best model system for nanomaterial safety assessment.

## 5. Advanced Models for In Vitro Testing

### 5.1. 3D Cultures

Although animal studies are considered physiologically relevant, their limited predictability, longer experimentation times, high costs, lack of high-throughput screening associated with the implementation of the 3Rs (reduction, replacement, and refinement) principle, and novel regulations that ban animal experimentation (e.g., in the cosmetics field) led to the need of developing advanced in vitro models [149,150,151,152]. So, worldwide research groups are dedicating efforts to develop a new generation of advanced in vitro models capable of recapitulating organ functions becoming effective tools for toxicology, pharmacology (investigating drug metabolism, pharmacokinetics, and toxicity), and for the mechanistic understanding of organ physiology and pathophysiology [150,153,154,155]. Several promising advanced models have been already established with superior physiological relevance, correlation, and validation compared to in vivo models [149,150,151,153,154,156,157,158]. However, it is important to stress that the priority in model development was not for nanosafety purposes.

Nanomaterial safety assessment starts traditionally with in vitro cultures using 2D systems that cannot mimic events observed in 3D structures such as native tissues [152]. Actually, the significant biological barriers and organs for nanotoxicological studies are the skin for dermal exposure, the gastrointestinal tract for oral uptake, the lungs (bronchial and alveolar epithelium) for inhalation, and the endothelium for intravenous exposure, while liver, lung, kidney, bone marrow, and spleen are important organs for studying nanomaterial accumulation [149,150,152]. It is widely reported that 3D models provide a closer and more realistic in vivo-like approximation that is more sensitive in identifying cellular responses to nanomaterials, wherein information on barrier penetration and translocation capabilities can be obtained [150]. In general, 2D cell monolayers usually overestimate the extent of nanotoxicity, whereas the 3D models provide a closer and more realistic in vivo-like approximation. An interesting example is skin organotypic 3D models that are already commercially available (e.g., EpiDerm, epiCS, EpiSkin, and SkinEthic) in which several Organisation for Economic Co-operation and Development (OECD) guidelines are employed to evaluate chemical safety. These seem to be suited to the assessment of skin-related nanomaterial risk [159]. Three-dimensional skin models exposed to nanomaterials provided more realistic analyses with lesser nanomaterial penetration due to an enhanced barrier function [158,159]. Oral epithelium and urogenital tract tissues are already commercially available; however, few companies provide relevant essential organs for nanomaterial safety such as the liver, kidney, respiratory epithelium, and intestinal epithelium [149,150,151,152,153,160]. An amazing effort was completed in a European Union (EU) project that was fully dedicated to the development of reliable and robust physiologically anchored tools for nanomaterial hazard assessment (PATROLS). The developed advanced models included the lung (coculture comprising immune cells, interstitial cells, or barrier cell types using permeable transwell membrane inserts), intestine (cell triple coculture differentiated epithelial and immunocompetent macrophage-like cell line), and liver (hepatocyte-based microtissue and a HepG2 human hepatocyte spheroid) that were specially fabricated to allow cytotoxicity, genotoxicity, and inflammatory responses to nanomaterials in more realistic exposure scenarios [161].

Spheroid and organoid technology have already started to contribute to nanotoxicity assessment, seemingly filling the gap between 2D and in vivo models and demonstrating a good predictive value and significant data correlation with clinical trials (in the case of nanomaterials for therapy) [150]. Liver, kidney, brain, skin, lung, and intestinal organoids among others were already established, and they are also considered promising tools for predicting nanomaterial toxicity to organs [150]. They provide organotypic cytoarchitectures with a spatially organized structure and in vivo phenotypic and extracellular matrix (ECM) expression mimicking important cellular functions such as cell migration, differentiation, and apoptosis, among other advantages [158]. In nanotoxicological studies, organoids seem to offer a barrier to nanomaterial distribution and cytotoxicity [9,152]. Lately, attention has been paid to the development of patient-specific and disease-realistic models that illustrate vulnerable individuals within populations. Excitingly, it was observed that the diseased hepatic spheroid model is more sensitive to nanomaterial toxicological stress than the equivalent healthy models [162]. Alternatively, 3D biomaterial-based models also present multiple benefits, since they are constituted by synthetic or naturally derived biomaterials, mimicking cell–ECM interactions [156,163,164]. With the latest improvements in materials science, microfabrication techniques, and bioreactor-based spheroid and organoid technology [153], new efforts are being made to combine self-organizing 3D biological structures into organs-on-chips (OoCs) to emulate both the structural and dynamic complexity of tissues and organs [153,158,165,166,167].

### 5.2. Organs-on-Chips

OoCs are one of the top ten emerging technologies considered revolutionary tools that may substitute animal experimentation in the future [153,154,167]. In recent years, microfluidic devices have demonstrated tremendous potential for developing in vivo-like cellular or tissue structures on a chip that may be leveraged for examining the safety assessment of nanomaterials in highly dynamic conditions. OoCs are normally designed using a reductionist approach (focus on key cellular constituents, structural organization, and biochemical and/or mechanical cues of the basic anatomical element responsible for organ/tissue function) with the versatility to increase the complexity of the system [166]. They employ bioengineering technologies to organize cells into “tissues” and facilitate fluid flow with the goal of constructing miniaturized tissue/organ testing devices that try to mimic the human body in all aspects (e.g., function, metabolism, architecture) [154]. They provide a 3D microenvironment that can be constructed with the aid of biocompatible materials (support cell growth) that together with biomechanical and biochemical cues and occasionally electrical signals are synchronized to model in vivo-like responses [155]. OoCs began to emerge at the turn of the century when Ingber et al. demonstrated the essential elements for lung organotypic function, showing that nanomaterial uptake from the air interface only occurred with the application of cyclic stretch [168]. A decade later, great advancements were seen in OoC models as a result of the major developments in microfabrication technologies, sensors, imaging, and biology. Currently available OoC examples are the brain, heart, lungs, liver, gut, pancreas, kidneys, skeletal muscle, adipose tissue, skin, cornea, cervix, amnion, placenta, blood vessels, and bone [155]. Many of these models comprise two overlapping perfusable channels separated by a polymeric permeable membrane, which permits the culture of two or more cell types in fluidically independent chambers [166].

Although significant advances were already observed in the past decades, there is still a significant lack of using OoCs to understand key biological mechanisms related to nanomaterial exposure and uptake, with most studies remaining in the proof-of-principle stage (Table 4). OoC examples of skin, lung, and gut are being used to test nanomaterial entrance; liver and kidney are being used to test metabolism and clearance; and bone marrow, blood vessels, and spleen are being used as toxicity-susceptible organs [153]. Microfluidic-based systems seem to be a versatile tool allowing the recreation of physiologically relevant measurement conditions, and more relevant to the nonstop monitoring of adverse effects of nanomaterials, they allow maintaining stable suspensions during cell exposure [151]. Scientific evidence suggests that biochemical and biophysical (e.g., stretch/strain forces for actuated tissues, or hemodynamic shear forces for vascular tissues) [165] cues formed in the complex biological microenvironment can have a profound impact on the comportment of nanomaterials, and these cues are not considered in most of the conventional 2D in vitro models [153].

Modulating the fluid flow in the model by a microfluidic device can assist in enhancing the bioavailability of the nanomaterials to the cells and leaving little opportunity for off-target nanomaterial deposition since localized nanomaterial deposition induces ROS production and inflammatory response leading to toxicity [166]. Although the presence of flow plays a pivotal role in nanomaterial toxicity in the human body, the interaction of nanomaterials with cells and their uptake and translocation under continuous perfusion is not clear and is even controversial [167]. The high variety of nanomaterials, the lack of standardization in nanomaterial preparation and dosage control, and alterations in the volume of the cell culture media and cell number cause the controversial and significant discrepancies in nanomaterial safety assessment in microphysiological models [151]. Efforts should be made to improve in vitro oriented simulation and optimize fluidic design considering the physicochemical properties of nanomaterials [151,153,167]. Another challenge is the small sizes of the chip associated with nanomaterials’ surface reactivity which can lead to nanomaterial surface adsorption on the small size of the channels. Nanomaterial absorption into the device can result in a misrepresentation of nanomaterial toxicity due to the reduction in nanomaterial concentration and consequent concentration–response interpretation. Alternative polymeric materials or chemical modifications of the chip surface are being proposed to minimize nanomaterial adsorption [167].

### 5.3. Multiple-Organs-on-Chips

Nowadays, technological innovations are aiding the scientific community in developing multiple-organs-on-chip (MOoC) devices. These are constituted by multiple tissues fabricated using two or more tissue chips or by incorporating multiple interconnected chambers representative of tissues/organs on one chip to recapitulate communications among different tissues [153]. Integrated multiorganoid models have been proposed in an effort to mimic complex procedures of metabolism and responses at the multiorganoid level, where flow rates resemble blood circulation in the human body [176]. Therefore, MOoCs can be exploited to evaluate the systemic toxicity of nanomaterials from the dynamic process of distribution, absorption, metabolism, and excretion features; nonetheless, systemic predictions are still challenging [157,158]. Future studies should take advantage of the development of engineered perfusable vasculature in microphysiological platforms making it possible to mimic the delivery of nanomaterials in a more physiologically relevant way, allowing the accurate prediction of their performance in vivo [153,166]. We believe that in the near future these platforms will provide new insights not only into systemic nanomaterial effects on different organs but also into their metabolites and subsequent secondary toxicity.

Besides many other advantages, MOoC models might allow nanotoxicity signatures at metabolite, protein, or gene levels that can be explored by combining them with diverse cutting-edge analytical approaches (e.g., fluorescence methods, microfluidics, artificial intelligence, multiomics, and single-cell analyses) [158]. The capability to integrate parallel streams on the same chip will allow high-throughput screening and will decrease the quantities of analytes needed and the overall testing time. Efforts should also be made to explore how single-cell analysis and bioinformatics tools can improve the mechanistic knowledge of adverse biological responses to nanomaterials in the physiological context of OoC and MOoC models.

### 5.4. Sensor Integration with Microphysiological Models

The advancement of the in vitro models used for nanotoxicology calls for the monitorization of important parameters related to the organ that is being emulated. Most approaches explored so far rely on off-chip and endpoint measurements and imaging; therefore, they do not take full advantage of all the possibilities offered by OoC platforms [177]. The integration of sensing strategies to enable real-time and in situ analysis of biological molecules and continuous detection of cellular functional changes boosts the utility of OoC systems. Additionally, integrated sensing technologies offer label-free, noninvasive approaches, with multiplexing, customization, and automation potential.

While most of the integrated sensing OoC platforms have been developed for the purpose of physiology or disease modeling (including cancer) and drug testing, their application to (nano)toxicology is possible and foreseen. So far, the focus has been mostly placed on the integrated monitoring of the microenvironment conditions of the OoC, such as temperature, pH, and oxygen levels, as well as on the establishment of a cohesive barrier function (transepithelial electric resistance (TEER)) (Table 4) [169,172,174,175,178,179,180]. Interestingly, HT measurements of some of these parameters have recently been achieved [181]. The analytical techniques mostly employed for these purposes are electrical (e.g., for TEER), electrochemical (e.g., for oxygen and pH), and optical (e.g., also for oxygen and pH), even though the latter ones rely on the use of indicator dyes [182]. The study of the metabolic function and the real-time release of tissue biomarkers in OoCs provides evidence of the tissue dynamics and their functional competence and maturity. The monitorization of some parameters, such as glucose levels and lactate production, supplies useful information regardless of the organ under analysis [183,184,185]. On the other hand, for certain tissues, the secretion of specific biomarkers benchmarks their utility for specific ends. For example, the online monitoring of transforming growth factor beta (TGF-β) production has shed light on the role of the communication between hepatocytes and stellate cells in the onset of liver injury [186]. Likewise, the continuous tracking of albumin, glutathione-S-transferase, and creatine kinase myocardial band production from a human heart-liver-on-chip platform shows great potential for understanding the crosstalk between both organs [187]. More recently, measuring insulin secretion in a pancreatic islet-on-a-chip in situ supports the application of the device for diabetes-related research [188]. These are representative examples of the infinite possibilities offered by OoCs with integrated sensing abilities, which are expected to cover any secreted biomarkers of relevance in the near future. Within the (nano)toxicology field, extracellular biomarkers of adversity can be targeted and tracked in real time, preferably in medium- to high-throughput formats. Their continuous footmark can help unravel the mechanisms by which nanotoxicity is triggered and propagated. Thus far, most of these advanced human-based sensing devices have been largely put to the test with drugs, so it seems a good time to start challenging them with nanomaterials.

## 6. In Silico Tools in Nanotoxicology

As has been previously mentioned, nanotoxicology and engineered nanomaterial production have seen rapid advancements and innovation over the last two decades. Despite the continuous effort from the scientific community, understanding the potential environmental and human health hazards of engineered nanomaterials is still a crucial challenge. In this regard, one of the current research frontiers in nanosafety is related to the use of in silico tools, such as quantitative structure–activity relationship (QSAR) models and the closely related quantitative property–activity relationships (QSPR) models. Such research interest is explained by the ability of QSAR/QSPR models to determine the relationship between an endpoint/target variable, i.e., biological activity or property, and some relevant structural characteristics (designated as descriptors) of the system.

Overall, the classical QSAR/QSPR workflow concerns (i) compilation of data from public databases or experimental procedures, (ii) data curation, (iii) descriptor calculation, (iv) model construction, and (v) statistical measures [189]. As in other fields related to nanosafety, the development of QSAR/QSPR models has been guided by a trade-off between predictive performance and interpretability. On one side, the model can comprise trivial molecular descriptors, such as physicochemical properties, and present a simple learning method such as linear regression [190]. On the other side, the model can contain continuous and data-driven molecular descriptors, which offer high predictive power and present complex black-box approaches such as artificial neural networks [191]. Currently, it is argued that a simple and informative model is more valuable to experimentalists [189].

Given that the developed models have been mainly applied to small molecules and traditional materials, the application of this process to nanomaterials requires adapting the methodologies to a higher level of complexity. One of the possible avenues to fulfill this requirement is applying new computational representations that include the necessary information to describe the structure and composition of such complex systems. In addition, as it is realistically impossible to test every nanomaterial experimentally, it is paramount to pave the way to a more predictive nanotoxicology. Therefore, Table 5 summarizes the most recent nanotoxicology predictive models focusing on the impact of nanomaterials’ properties, structure, and composition on their cytotoxicity and inflammatory potential. As a complement, a dynamic visual representation (Figure 7) was obtained using GEPHI (software version 0.9.2). The detailed pipeline to design the network is described elsewhere [192]. Overall, the network focuses on (i) the type of nanomaterial, (ii) descriptors, and (iii) the QSAR model used to predict a defined endpoint.

As a first interpretation of the network (Figure 7a), it is possible to identify that metal oxide, metal, and carbon-based nanomaterials are currently the most studied. Regarding the descriptors, the most appealing are the physicochemical properties (such as molecular weight and octanol–water partition coefficient), the optimal quasi-SMILES (i.e., a sequence of symbols that represent the nanomaterials by their properties and the experimental conditions involved in their experiments), and the structural information (namely the electronegativity of each atom). Concerning the models, the complexity varies from linear regression to an artificial neural network. Nonetheless, the most used models are Monte Carlo (MC), multiple linear regression (MLR), and partial least squares (PLS) regression.

By taking advantage of the interactivity of the network, it is possible to highlight a specific class and understand the existing connections between the nanomaterial type, the descriptors, and the model used to predict a defined endpoint. Hence, Figure 7b highlights the metal oxide class and shows that several descriptors successfully represented such nanomaterials. Moreover, it identifies that this information set allowed training and implementing several QSAR predictive models, going from linear regression to tree-based ensemble algorithms such as decision trees and random forest.

As a representative example, Roy et al. [195] used the second-generation periodic table-based descriptors to represent metal oxide NPs, such as TiO_2_ and ZnO. Ultimately, an MLR model was trained and used to predict the cytotoxicity of the nanoparticles on *Escherichia coli* under different conditions, achieving an R^2^ of ≈ 0.77. Additionally, Appendix A highlight the metal and carbon-based nanomaterials, respectively.

As mentioned above, one of the exciting approaches for representing nanomaterials is the use of quasi-SMILES. Such research interest is explained by the ability of quasi-SMILES to encode the structure, composition, and physicochemical properties of a specific nanomaterial. Moreover, if reasonable, quasi-SMILES can also encode the experimental conditions involved in the nanomaterial experiment. Thus, quasi-SMILES has become a simple but powerful tool to represent nanomaterials to QSAR approaches. In this regard, Figure 7c highlights the optimal quasi-smiles-based descriptors and the closely related MC algorithm, showing that quasi-SMILES has been used to represent metal oxide, metal, and carbon-based nanomaterials. As a representative example, Toropova et al. [204] developed an MC model to predict the cellular uptake of gold NPs by A549 lung carcinoma cells. To do so, gold NPs were represented by quasi-SMILES that considered the size of the NPs, the first and second ligands interacting with the NPs, and the cellular uptake in A549 cells. Then, these optimal descriptors were collected and used to train an MC model, achieving an R^2^ of 0.7–0.9.

In summary, the most recent works covering predictive nanotoxicology and different representations of relevant structural characteristics of nanomaterials have been briefly presented and discussed in this section. Interestingly, it is noticeable that most of the works focus their attention on nanoparticles. Despite being a recent frontier in nanosafety, the in silico tools have been increasing in predictive efficiency for specific problems and consequently increasing in popularity among the scientific community. Nonetheless, achieving a broader predictive nanotoxicology, where an in silico tool can be applied to different problems with similar efficiency, is still a crucial challenge. To overcome such a challenge, the scientific community needs to reinforce the relationship between experimentalists and computational scientists and to start or continue implementing the findability, accessibility, interoperability, and reusability (FAIR) data principles [212]. Currently, there are interfaces, such as eNanoMapper [213], that provide and collect FAIR data to support nanosafety assays. Following these principles and using the data collection templates makes it possible to collect experimental data standardly. Consequently, the amount of data would exist in greater quantity and diversity, which would allow the development of in silico tools with a broader domain of applicability.

## 7. Life Cycle Assessment and Nanosafety

The current paradigm of the nanotoxicology, which relies mainly on in vitro and in silico models, is aiming to develop safe nanomaterials at an early stage of the innovation process in order to minimize the potential environmental and human health hazards. Therefore, as previously stated, it is imperative to ensure that the generated data are FAIR, so they can be used in an efficient way, especially bearing in mind the current expectations for safer and sustainable emerging technologies. This transition is crucial, and it has been rising in the last few years, as a result of the ambitious roadmap of the recent EU policy initiatives and the sustainable development goals of the United Nations. The plan involves a recognized demand for a competitive system for the efficient use of resources and energy, while minimizing emissions and waste production, simultaneously promoting a circular economy [214,215,216].

Nanotechnology is an emerging field promising innovative solutions in a wide range of applications that are expected to greatly contribute to sustainable development [217]. Nevertheless, the fast growth of nanotechnology research associated with the increased interest in novel systems and nanomaterials has not been completely followed by sufficient knowledge about safety and sustainability issues [218,219,220,221,222]. Hence, over the last few years, the scientific community from the diverse fields of the nanosafety framework has made an effort to focus research on identifying and understanding the potential risks and impacts that this emerging field may have on humans and the environment [221], as has been pointed out throughout this review. For instance, and as an outcome example, a series of tools has been developed under the scope of different European projects (e.g., NanoFase, NanoReg2, SUNSHINE, SbD4Nano, ASINA, and SAbyNA), some of which are still ongoing. Most of these tools are based on a life cycle concept, in which the idea is to establish an optimized safety strategy, preferentially starting at an early stage of the nanotechnology innovation process development [223,224,225]. Additionally, the new framework intends not only to guarantee safety but also to ensure sustainability concerns are addressed [226]. The plan is, therefore, to maximize or at least maintain the functionality and the lifetime of nanomaterials and nano-enabled products in a circular system, together with the efficient design of the production process that ensures the minimization of the environmental impacts while guaranteeing the safety across their life cycle [217,227]. In this sense, the LCA is a powerful tool that can be applied in a wide range of areas as a cross-cutting approach, providing a comprehensive understanding of the potential environmental impacts while identifying the main hotspots for a potential improvement and ensuring the environmental sustainability of such products and processes. The LCA is an international standardized methodology comprising four main phases (Figure 8), as defined in the ISO 14040 series: (i) goal and scope definition, (ii) life cycle inventory (LCI), (iii) life cycle impact assessment (LCIA), and (iv) interpretation.

The idea is to use the LCA along with the risk assessment perspectives to integrate both safety and environmental issues with functionality and socio-economic indicators (e.g., Social LCA (S-LCA) and Life Cycle Cost (LCC)) into a decision support system, such as the so-called Safe and Sustainable-by-Design (SSbD) approach [217]. The general concept is basically a multiobjective optimization process, and a novel integration approach for decision-making was recently developed under the scope of the ASINA project through mathematical algorithms solving a multicriteria decision analysis (MCDA) model [228]. Nevertheless, even with the advances in the SSbD approach, there are still challenges in determining how to properly provide strategic guidelines regarding safety and sustainability issues among all the stakeholders, from the nanotechnology innovators to the regulatory policy-makers [225,229,230].

A crucial factor is essentially related to the complexity and dynamic behavior of the nanomaterials which then leads, as was previously stated, to a lack of tools and proficient measurements of quantitative and qualitative data. The ongoing development of analytical techniques to detect and characterize nanomaterials is even more critical in the case of complex natural systems [231], especially due to the problems in the differentiation between natural and engineered nanomaterials [232,233].

The releases of nanomaterials can occur during any stage of the life cycle of a nano-enabled product, and, for instance, humans can be exposed through different exposure routes (e.g., inhalation, dermal, and ingestion) linked to a wide range of situations (e.g., occupational workplaces, medical treatments, use or consumption) [234]. Therefore, it is essential to understand not only those releases of the nanomaterials connected to the pristine form but also the ones from the further transformations that may occur due to the aging and/or altered processes [233,235,236], as well as the probable interactions with the surrounding setting, which could affect the potential fate and toxicity effect [237]. Thus, a holistic analysis should be considered, comprising aspects from the extraction to the end-of-life management scenarios; therefore, the LCA has been considered as an appropriate tool to use [238,239].

In the last two decades, a number of studies related to the LCA application in a wide range of nanotechnology fields, using different types of nanomaterials, have been conducted in a total of 128 works (Figure 9). The reference studies are from 2001 to 2022 and were mostly recompiled from a few review studies [240,241,242] and from [222,243,244,245,246,247,248,249,250,251,252,253,254,255]. From this analysis, it is possible to observe a wide variety of nanomaterials and application fields. The main studied nanomaterials were TiO_2_ and CNTs, including single-wall CNTs and multiwall CNTs, while the synthesis and production were the major fields requested to be assessed through the LCA.

Although many studies have been published in the last few years, there are still many limitations, as there were in the beginnings of LCA application [238]. This has been stated in recent reviews [240,242], and it was also debated in an LCA Discussion Forum a few years ago [256]. Some of those limitations are related to the system boundary considerations. The most comprehensive ones are the cradle-to-grave and cradle-to-cradle analysis, which comprise all the life cycle stages of a system—the raw material extraction, production, transport, use, and end-of-life management, including recycling/upcycling. However, a critical problem is that most of the existing LCA studies do not consider the end-of-life management scenarios (e.g., wastewater treatment plants (WWTPs), incineration, and landfill) that could act as sink compartments where the nanomaterials potentially accumulate [19]. Actually, to the best of our knowledge, there is no international regulation on the disposal management of nanomaterials [242]. This could be a concern accounting for the increased availability of nano-enabled products which will eventually enter into various waste treatment processes with no specific treatment being considered [257]. Taking into account the wide array of product categories in which nanomaterials could be present, such as plastics, inevitable exposure pathways to nanomaterials through the waste management scenarios are foreseeable. For instance, this could occur through the incineration of nonrecyclable fractions and, consequently, the landfilling of the produced ashes [258]. In the case of those recyclable fractions, the persistence of nanomaterials could represent a restriction in terms of safety reuse, which could be a problem when considering the need to maintain a circular economy. Furthermore, other important exposure pathways are linked to the increased use of the personal care products—containing nanomaterials—that could enter directly into the water compartment (freshwater and marine water) and, indirectly, through the WWTPs, from there probably reaching the soil compartment through the sludge valorization [258,259]. Previous studies stated that the waste treatment of those products containing large amounts of carbon-based nanomaterials could be eliminated during incineration and in the WWTPs, whereas a persistent potential was observed for nanomaterials containing metals and metal oxides [260]. In addition, some efforts focusing on the detection and characterization of nanomaterials in waste streams have been made [261,262], but more studies are needed to elucidate the potential risk of waste management treatments.

Another important constraint is the lack of LCI datasets, even those not related to the nanomaterials. Most of the available data could be obtained from the literature, databases, and lab and pilot-scale studies, but the information is still generally limited [263]. The accuracy of the data collected has an important effect on the reliability of the results, and the use of primary data is also recommended. In addition, for emerging technologies that are not yet established at an industrial scale, the potential impacts—mainly in terms of energy and materials supplies—could be overestimated, considering that the optimization is usually attained with the upscaling of such processes [263]. Therefore, the involvement of the manufacturing industries is fundamental for those well-established processes. Nevertheless, the scarcity of exchange of information from the industry sector has been identified as a general problem due to confidentiality issues and is also associated with the difficulty of tracking the final commercial application of the nanomaterials. Hence, the need for a transparent exchange of information among all the stakeholders is urgently needed.

An additional limitation relates to the assessment of the potential impacts, especially those on human health and the environment, triggered by the released nanomaterials. This is because there is no consensus on how to approach appropriate modeling in the LCA framework, and there is a challenge in determining how to account for those releases in the LCI modeling. For instance, the Material Flow Analysis (MFA) model has been used for tracking the flows of nanomaterials across their entire possible life cycle, from the technosphere to their relocation into several compartments of the environment [264,265]. Some recent improvements have been attained, especially those related to the inclusion of different sizes and forms of nanomaterials in the dynamic modeling [266,267]. This could be an important advance for the incorporation of nanomaterial releases in the LCI modeling, considering the recommendation of having specific features of the nanomaterials (e.g., shape, size, crystalline structure, surface charge, and surface area), in addition to mass and chemical composition, which are the ones considered for conventional chemicals [268]. This approach will be an essential step forward for tracking the releases of nanomaterials across the life cycle of nano-enabled products considering that, for instance, the loss or gain of surface functional groups will determine their environmental fate, exposure, and toxicity potential [269].

Another critical challenge, also related to the possible impacts on human health and the environment linked to the releases of nanomaterials, is the lack of characterization factors (CFs). In the LCIA, different models are used to translate the quantity of each emission in order to evaluate the environmental impacts through multiple impact categories (e.g., human toxicity, global warming potential, and resource depletion). Therefore, the emissions are converted into environmental damages through three different models (fate, exposure, and effect) to obtain a specific CF. This requires both a qualitative and quantitative assessment along with a mechanistic understanding of each of these components. From the analysis of the previous LCA studies in the field of nanotechnology, it is evident that there is a lack of information related to the inclusion of those impacts caused by the releases of nanomaterials, due to the absence of specific CFs (Figure 10).

To characterize the toxicity impacts that are required in the LCA calculations, the OECD guidelines recommend the use of the USEtox model once it is appropriate to apply it to nanomaterials [270]. In fact, and although the USEtox model was firstly oriented to chemicals, some efforts have been made in order to attempt the establishment of CFs for specific nanomaterials, including the most suitable one so far for the specific case of TiO_2_ [271]. Within the USEtox framework, apart from the fate and exposure factors, the effect factors to account for the particular case of freshwater ecotoxicity impact are calculated through the fraction of species (representative of the three trophic levels—algae, crustacean, and fish) that are potentially affected by the exposure to a specific substance. The recommendation for nanomaterials is to apply an approach based on the geometric mean at the trophic level, rather than the species level, in order to provide a proper distribution of the toxicity data [240]. On the other hand, for the human toxicity assessment, the effect factor reflects the potential to increase human disease due to the change in lifetime intake of a particular substance, extrapolated from animal in vivo models [272]. This could also be another critical point—in a good way—considering the new paradigm for toxicology, wherein there is a need to replace animal testing with alternative in vitro models, as previously stated in this review. This transition could solve some problems with the use of animal data during the LCIA [270]. In fact, a recent method to determine human toxicological effect factors for some nanomaterials, by using in vitro data, was proposed [273]. Nevertheless, more strategies are needed for comparison purposes.

Despite all the progress, there is still no agreement on which specific nano fate descriptors (e.g., agglomeration, aggregation, and sedimentation), physicochemical properties, and particle characteristics, among other features, need to be integrated into specific mechanistic models [274]. This is challenging, especially considering the amount and the wide variety of nanomaterials that should be studied; therefore, the use of in silico analysis will provide crucial information on the relevant characteristics of the exposure related to a particular or to a group of effects, as discussed in the previous section. Therefore, further improvements in this topic are needed, considering that some nanomaterials, including their forms, probably cannot be clustered and read across due to their specificity in terms of both environmental release profile (i.e., amounts, form, and compartment) and adverse effects [275].

Noteworthily, and considering that it is even more imperative to determine the environmental impacts of the emerging technologies at an initial stage of development, some shortcomings are expectable during the assessment [263,276]. Nevertheless, even with all the challenges and uncertainties, a prospective LCA should be performed in a useful way that could lead to a decisive strategy to orient the investments and predict the upcoming environmental impacts [277], such as those linked to the critical raw materials, in order to ensure the sustainability of such technologies.

## 8. Conclusions

Nanosafety is a very broad concept related to many different approaches for evaluating the potential deleterious effects of nanomaterials to reduce the potential impact on the environment and health. For this, traditional methods of toxicology (i.e., cytotoxicity, immunotoxicity, genotoxicity) have been adapted to evaluate nanomaterials. New approaches using high-throughput methods allowing the evaluation of multiple outcomes in a single sample are also part of the new trends aiming to tackle the rapid growth of the nanomaterials field. In addition, new technologies, such as OoCs and MOoCs, have emerged as new platforms for predicting drug efficacy, but the scientific development for nanosafety assessment is still delayed. We consider that the expansion of OoCs with novel development in fields such as bioimaging will improve label-free analysis on chips, bringing information at the tissue and cellular levels. Innovative detection methods and analysis possibilities will help to accurately study cellular responses and communication among tissues or organs, prolonging culture timeframes, to better detect the long-term performance and predictability of nanomaterial toxicity. All these adaptations of traditional and new methods are critical for feeding the in silico models aiming to predict the effects of nanomaterials, which are becoming critical for the Safe by Design concept. Finally, going beyond the effects of nanomaterials in living organisms, the LCA assessment must be performed to have a clear image of how the production, use, and disposal of nanomaterials may have an impact on the sustainable use of nanomaterials by industry.

## Figures and Tables

**Figure 1 nanomaterials-12-01810-f001:**
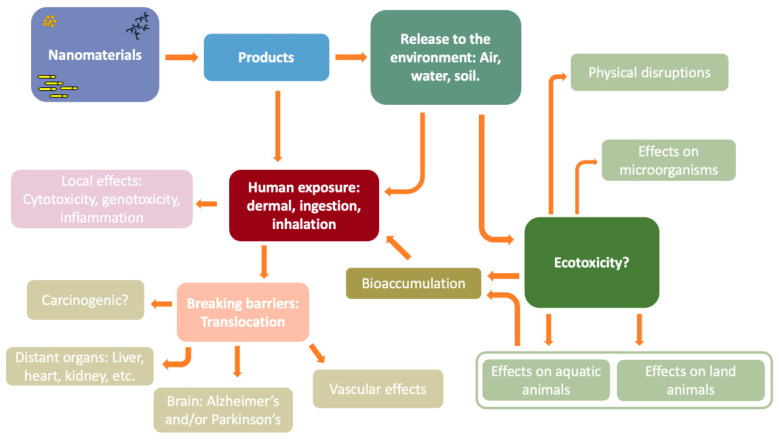
General view of possible interactions, routes of exposure, and adverse outcomes that can be triggered by exposure of humans and the environment to nanomaterials.

**Figure 2 nanomaterials-12-01810-f002:**
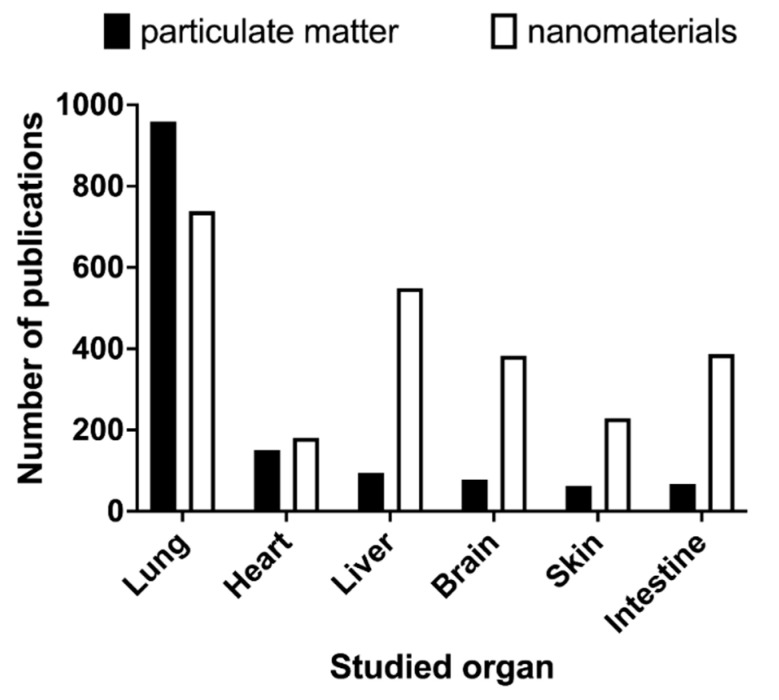
The main target organs studied in the context of (nano)toxicology. A PubMed search for “particulate matter toxicology” or “nanomaterial toxicology” followed by the organ identified on the X-axis was performed in April 2022. No other filters were applied. Of note, using the words “pulmonary”, “cardiovascular”, and “gastrointestinal” instead of the specific organ yields even higher numbers for the 3 categories, but lung-related toxicology still prevails by far.

**Figure 3 nanomaterials-12-01810-f003:**
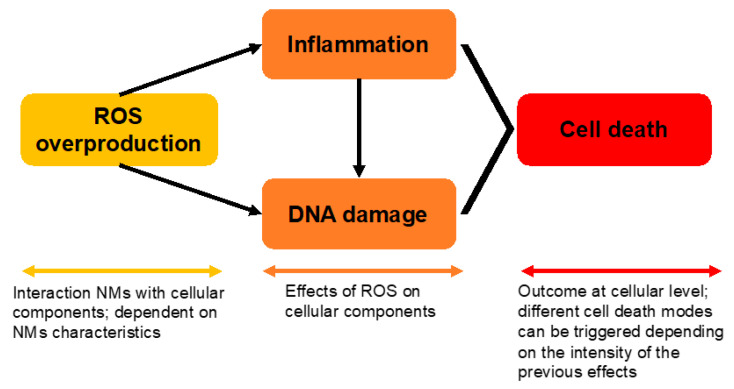
Predominant mechanisms of nanomaterial-induced toxicity identified so far and their presumed interaction.

**Figure 4 nanomaterials-12-01810-f004:**
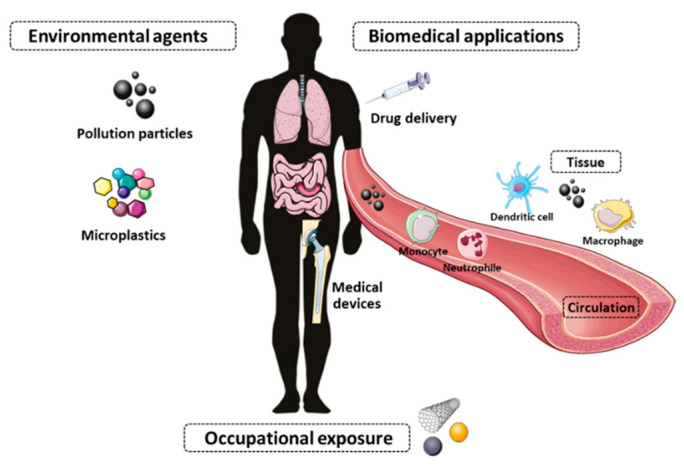
Exposure to nanomaterials activates the immune surveillance system. Nanomaterials used for industrial and biomedical applications or present in the environment can have a major impact on human, animal, and plant health. If nanoparticles penetrate anatomical barriers, cells of the innate immune system (e.g., macrophages, monocytes), found in circulation or locally in different tissues, recognize them. This may lead to nanoparticle degradation/elimination or modulate the body towards beneficial or detrimental responses. (Servier Medical Art, smart.servier.com).

**Figure 5 nanomaterials-12-01810-f005:**
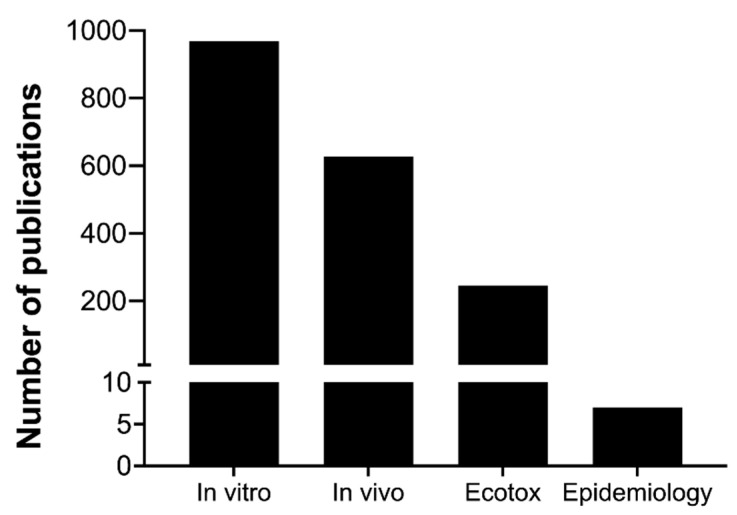
Nanomaterial-induced genotoxicity in various model systems (the figure is generated with the numbers of published papers appearing in PubMed database with specific keyword search; epidemiology mainly represents “occupational exposure”-related studies; ecotoxicology model species include mainly fish species, drosophila, bivalve mollusks, *C. elegans*, white worms, yeast, etc.).

**Figure 6 nanomaterials-12-01810-f006:**
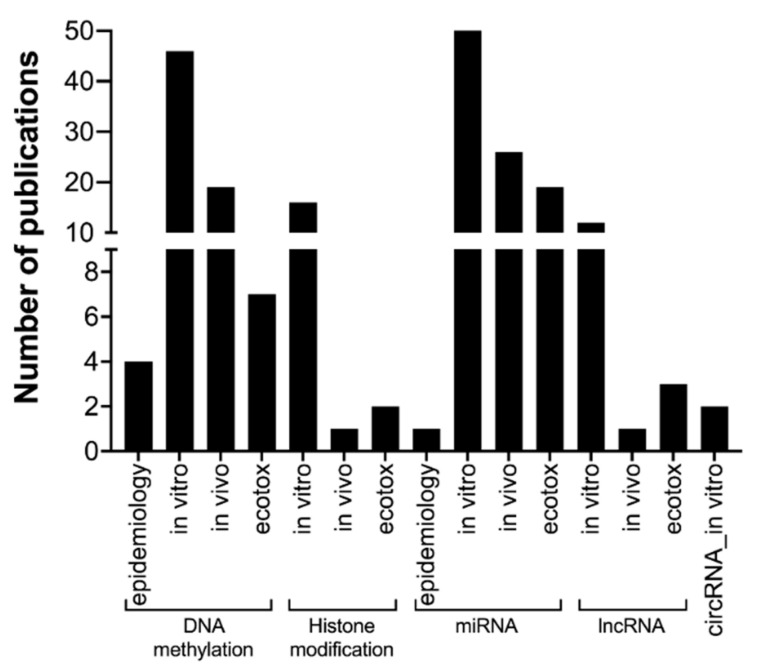
The nanomaterial-induced alterations in different epigenetic biomarkers based on various model systems (the figure is generated with the numbers of published papers appearing in the PubMed database with a specific keyword search; epidemiology mainly represents “occupational exposure”-related studies; in ecotoxicology, model species include mainly zebrafish, yeast, and *C. elegans*).

**Figure 7 nanomaterials-12-01810-f007:**
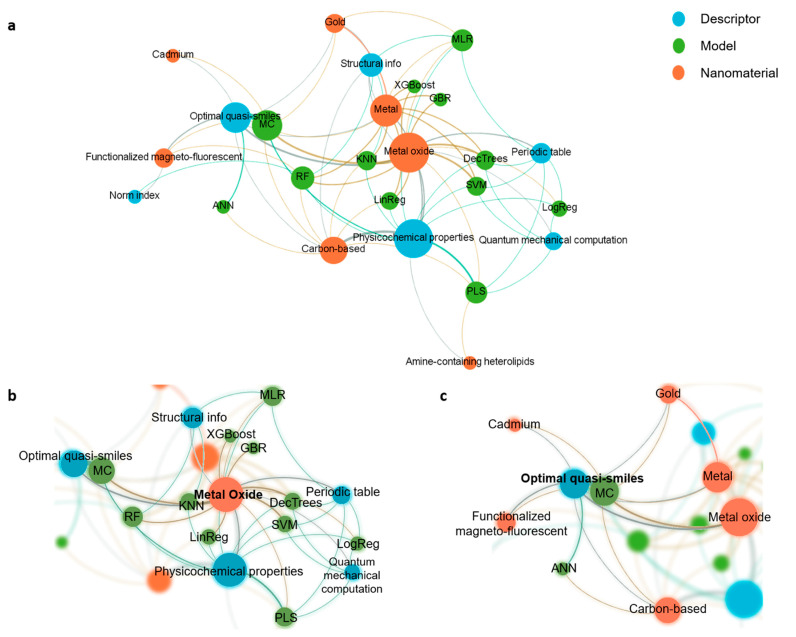
Graphical network of the 19 articles that meet the criteria to be included in Table 1. (**a**) Network with all the connections between the types of nanomaterials, their relevant structural characteristics (descriptors), and the QSAR models used to predict a defined endpoint. (**b**) Most relevant connections of metal oxide class. (**c**) Most relevant connections of optimal quasi-SMILES-based descriptors and the closely related Monte Carlo (MC) algorithm. ANN: artificial neural network; DecTrees: decision trees; GBR: gradient boosting regressor; KNN: k-nearest neighbors; LinReg: linear regression; LogReg: logistic regression; MC: Monte Carlo; MLR: multiple linear regression; PLS: partial least squares; RF: random forest; SVM: support vector machine; XGBoost: extreme gradient boosting.

**Figure 8 nanomaterials-12-01810-f008:**
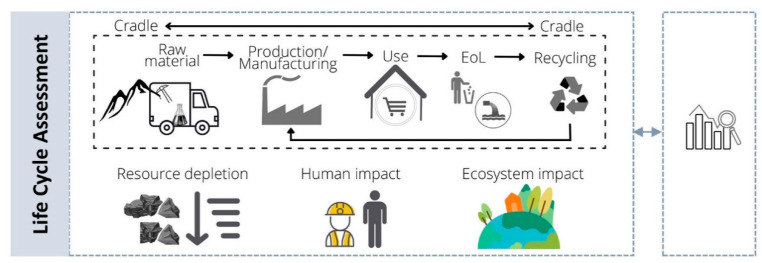
General conceptual framework of LCA representing the four main phases. Phase I: Cradle-to-cradle is one of the system boundaries that could be established in this stage; phase II: during the LCI, for each of the stages of the life cycle, all the inputs (e.g., natural resources) and outputs (e.g., emissions) within the system boundaries are accounted for; phase III in the LCIA, different models are used to translate the quantity of each emission in order to evaluate the potential impacts; phase IV: interpretation of the LCA results. EoL: end-of-life.

**Figure 9 nanomaterials-12-01810-f009:**
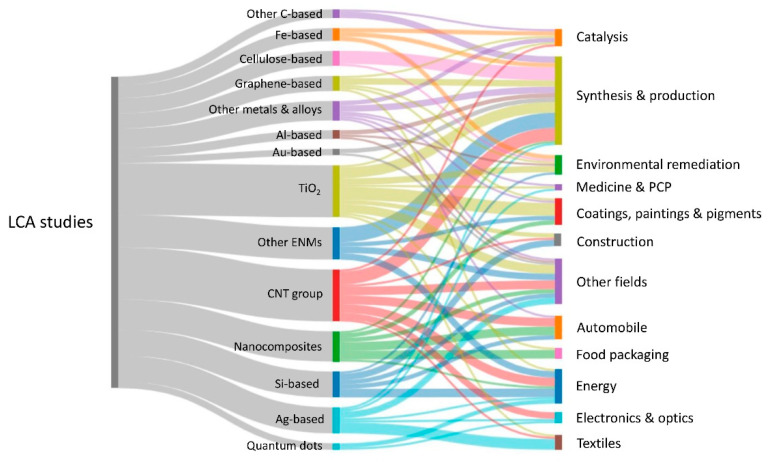
LCA application in different nanotechnology fields, including the nanomaterials studied. ENMs: engineered nanomaterials; PCP: personal care products; other metals and alloys: some of the most studied were Zn-based and Cu-based; other ENMs: some of the most studied were tungsten-based and zirconia-based.

**Figure 10 nanomaterials-12-01810-f010:**
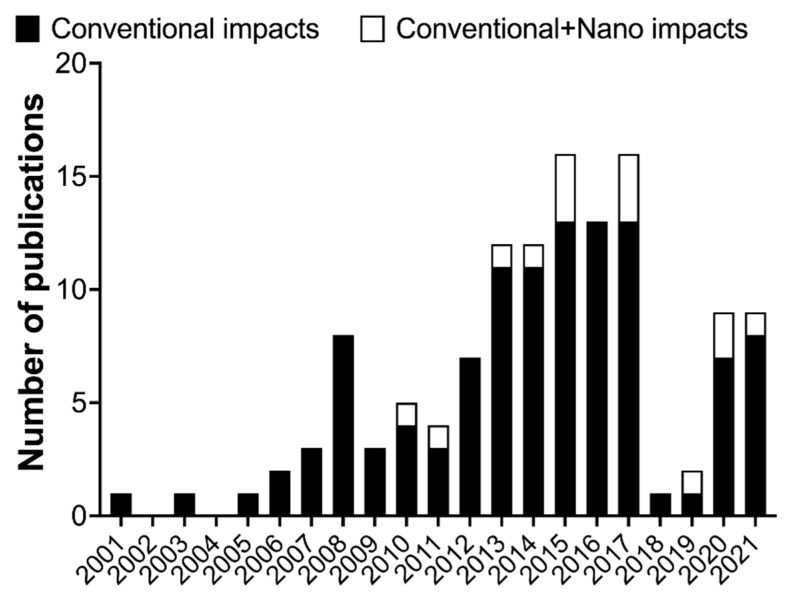
Number of publications related to the LCA in nanotechnology accounting for conventional impacts and also for those impacts linked to the releases of nanomaterials.

**Table 1 nanomaterials-12-01810-t001:** Main interferences from nanomaterials in some of the most widely used conventional toxicological assays identified so far.

Conventional Methodology	Observed Interference		Proposed Solution
	Cause	Result/Interpretation	
MTT reduction LDH leakage WST reduction	NM optical density; NM aggregation in cell medium	Falsely increased viability	Sample centrifugation after cell lysis
	NM redox activity	Falsely decreased viability	None
ELISA (cytokine release)	Protein adsorption to NMs	Falsely decreased cytokine production	Add serum proteins to NM suspension
Comet assay	Interference enzyme activity	Falsely decreased genotoxicity	None
ROS quantification (H_2_DCF-DA)	NM redox activity	Falsely increased ROS levels	None
	NMs quench fluorescence; NMs scatter emitted fluorescence	Falsely decreased ROS levels	Sample centrifugation after cell lysis

NM: nanomaterial; MTT: 3-(4,5-dimethylthiazol-2-yl)-2,5-diphenyltetrazolium bromide; LDH: lactate dehydrogenase; WST: water-soluble tetrazolium salts; ELISA: enzyme-linked immunosorbent assay; ROS: reactive oxygen species; H_2_DCF-DA: 2′,7′-dichlorodihydrofluorescein diacetate.

**Table 2 nanomaterials-12-01810-t002:** Current testing strategies for nanomaterial-induced genotoxicity assessment.

Genotoxicity Marker	Assays	References
Gene mutation	Bacterial reverse mutation (Ames test)	OECD TG 471
	In vitro mammalian mutagenicity assay: mouse lymphoma (L5178Y) TK+/-assay	OECD TG 490
	In vitro mammalian mutagenicity assay: HPRT assay	OECD TG 476
	In vivo gene mutation assay (transgenic rodent somatic and germ cell gene mutation)	OECD TG 488
Chromosomal damage assays	In vitro chromosomal aberration assay	OECD TG 473
	In vitro MN assay	OECD TG 487
	In vivo (mammalian bone marrow) chromosomal aberration test	OECD TG 475
	In vivo MN assay (mammalian erythrocyte MN)	OECD TG 474
DNA damage (strand-break and DNA adduct)	In vitro comet assay Modified in vitro comet assay with DNA repair enzymes (e.g., OGG1, FPG)	JaCVAM EURL-ECVAM/ICCVAM [94,99]
	In vivo (mammalian alkaline) comet assay	OECD TG 489
DNA damage (DNA adduct)	HPLC/MS; ELISA	[104,109,110]
DNA damage response and repair	The γH2AX and 53BP1 foci count assay	[108,111]
	Multiplex array for DNA repair activity	[109,110]
	FM-HCR assay	[112]

OECD TG: Organisation for Economic Co-operation and Development Test Guidelines; HPRT: hypoxanthine-guanine phosphoribosyltransferase; MN: micronucleus; JaCVAM: Japanese Center for the Validation of Alternative Methods; EURL-ECVAM: European Union Reference Laboratory for Alternatives to Animal Testing; ICCVAM: Interagency Coordinating Committee on the Validation of Alternative Methods; HPLC/MS: high-performance liquid chromatography–mass spectrometry; ELISA: enzyme-linked immunosorbent assay; FM-HCR: fluorescence multiplex−host-cell reactivation.

**Table 3 nanomaterials-12-01810-t003:** Common methodologies for epigenetic endpoints applied for nanomaterial studies.

Epigenetic Endpoints	Specific Epigenetic Markers	Analytical Methods	References
DNA methylation	Global DNA methylation screening (5mc, 5hmC, 6mA, etc.)	HPLC/MS, ELISA, methylation-sensitive comet assay, pyrosequencing (repetitive sequences LINE-1 or Alu)	[121,122,123,124]
	Gene-specific promoter methylation	Methylation-specific PCR	[121,125]
	Differentially methylated regions (whole-genome sequencing)	MPS, DNA methylation-specific microarrays, MeDIP followed by sequencing	[121,126]
Histone modification	Whole genome (specific histone marker)	ChIP with DNA microarray, ChIP-Seq, ChIP-Chip	[127,128]
	Gene-specific histone (specific) modification	ChIP-qPCR	[128,129]
	Global histone modification markers	HPLC/MS, ELISA, immunostaining, immunoblotting	[130,131]
Noncoding RNAs	Whole genome	RNA-seq, microarray	[132,133]
	Gene-specific	qPCR	[134]

DNA: deoxyribonucleic acid; HPLC/MS: high-performance liquid chromatography–mass spectrometry; ELISA: enzyme-linked immunosorbent assay; 5mC: 5-methylcytosine; 5hmC: 5-hydroxymethyl cytosine; 6mA: 6-adenine methylation; PCR: polymerase chain reaction; MPS: massively parallel DNA sequencing; MeDIP: methylated DNA immunoprecipitation; ChIP: chromatin immunoprecipitation (ChIP); qPCR: quantitative PCR; RNA: ribonucleic acid.

**Table 4 nanomaterials-12-01810-t004:** Single- and multiple-organ-on-chip models employed in nanomaterial safety assessment.

Advanced Cell Models	Cell Types	Nanomaterial Exposure Conditions	Sensorization	Toxicological Assays	Key Biological Outcomes
Heart microphysiological system	NRVMs	TiO_2_ NPs at 10 and 100 μg·mL^−1^ and Ag NPs at 50 μg·mL^−1^	Electrical sensors	LDH assay, MTT assay	The high-dose exposure of TiO_2_ NPs (100 μg·mL^−1^) demonstrated impaired contractile function and damaged tissue structure after 48 h of exposure. Ag NP exposure caused cytotoxicity [169]
Blood–brain barrier on a chip	HAs and HUVECs	INPM exposure at 0, 5, 10, 20, and 40 μg·mL^−1^	-	ROS detection assay, CCK8 assay	The INPM could potentially activate several inflammatory pathways that directly damage brain structures and further lead to neurological diseases [170]
Liver on a chip	PRHs	10 nm Fe_3_O_4_ NPs	-	-	Perfusion of Fe_3_O_4_ NPs results in the reduction in albumin and urea production, indicating potential liver injury [167]
Lung on a chip	HPAEpiC, HUVECs, and THP-1	PM2.5 exposure at 200 and 400 μg·mL^−1^	-	Immunofluorescence staining assay, FITC-dextran permeability assay, ELISA	A low concentration of PM2.5 causes limited cytotoxicity, but a higher concentration of PM2.5 (>200 μg·mL^−1^) could significantly increase the ROS generation, apoptosis, and inflammation responses of epithelial cells and endothelial cells on the barrier and attachments of monocytes to the vessels [171]
	BEAS-2B and HUVECs	CSE at 10, 20, and 50 μg·mL^−1^	-	RT-PCR, ELISA, Western blotting	Lung on a chip enables the study of nanoparticle adsorption during various breathing frequencies, puff profiles of smoking, breath-holding patterns during inhalation and exhalation, and particle deposition in the lungs and the respiratory tract [154]
Placenta barrier on a chip	BeWo	20 nm SiO_2_ and TiO_2_ NPs, and 80 nm ZnO NPs for 24 h	Membrane-bound impedance sensor array	ROS detection assay	SiO_2_ and TiO_2_ NPs induced no loss in barrier integrity. In contrast, ZnO NPs displayed severe acute cytotoxicity already after 4 h [172]
	BeWo and HUVECs	TiO_2_ NPs exposure at 50 and 200 μg·mL^−1^	-	Immunofluorescence staining assay, ROS detection assay	Gradually increased cell death with increasing concentrations of NPs, thereby potentially leading to placental membrane rupture [173]
Gut/liver on a chip	Caco-2, HT29-MTX + HepG2, C3A	50 nm carboxylated PS NPs	-	AST assay	Gut/liver chip model demonstrates compounding effects of interorgan crosstalk between gut and the liver in facilitating NP toxicity [167]
Lung/liver/kidney on a chip	A549 + HepG2 + TH-1	Ag, Au-PEG, TiO_2_, and SiO_2_-FITC NPs	TEER measurements	Live/dead assay	The interconnection of the different modules aims at the simulation of whole-body exposure and response. SiO_2_-FITC NPs showed a cytotoxic effect on TH-1 after 12 h, which could be due to the interaction of NPs with cancerous cells releasing a substance that may have induced a cytotoxic effect [174,175]

NRVMs: neonatal rat ventricular myocytes; TiO_2_: titanium dioxide; NPs: nanoparticles; MTT: 3-(4,5-dimethylthiazol-2-yl)-2,5-diphenyltetrazolium bromide; LDH: lactate dehydrogenase; HAs: human astrocytes; HUVECs: human umbilical vein endothelial cells; INPM: indoor nanoscale particulate matter; ROS: reactive oxygen species; PRHs: primary rat hepatocytes; CCK8: cell counting kit 8; Fe_3_O_4_: iron oxide; HPAEpiC: human alveolar epithelial cells; THP-1: human acute leukemia monocytic cells; PM2.5: fine inhalable particles, with diameters less than 2.5 µm; FITC: fluorescein-5-isothiocyanate; ELISA: enzyme-linked immunosorbent assay; BEAS-2B: human bronchial epithelial cells; CSE: cigarette smoke extract; RT-PCR: reverse transcription polymerase chain reaction; BeWo: human choriocarcinoma cells; SiO_2_: silicon dioxide; ZnO: zinc oxide; Caco-2: human colorectal adenocarcinoma cells; HT29-MTX: human colorectal adenocarcinoma cells with epithelial morphology; HepG2: human liver hepatoma cells; C3A: clonal derivative of HepG2; PS: polystyrene; AST: aspartate aminotransferase activity; PEG: polyethylene glycol; A549: adenocarcinoma human alveolar basal epithelial cells; TH-1: Type 1 T helper cells; TEER: transepithelial electrical resistance.

**Table 5 nanomaterials-12-01810-t005:** Overview of the most recent nanotoxicology predictive models. A PubMed search for “QSAR and nanoparticles” from 2020 to 2022 was performed in April 2022. No other filters were applied.

Nanomaterials	Descriptors	Models ^1^	Main Goal
FD	204 molecular descriptors generated from the QSAR analyzing tools of BIOVIA Discovery Studio	LinReg	Predict the physicochemical properties of FDs that promote their cytotoxic effects/anticancer activity [193]
Metal NPs	24 physicochemical descriptors and toxicity data	MLR	Predict the toxicity and design the structures of metal NPs with low toxicity [194]
Metal oxide NPs	61 periodic table descriptors	MLR	Predict and investigate the essential descriptors responsible for the cytotoxicity of metal oxide NPs on *E. coli* cells under different conditions [195]
Gold NPs	Structural information (i.e., Dragon descriptors) of the surface ligands	MLR	Predict possible relationships between the oxidative reactivity of gold NPs and their cytotoxicity [196]
Carbon NPs	Physicochemical descriptors (molecular weight, overall surface area, volume, specific surface area, and sum of degrees)	Orthogonal PLS regression	Predict the interaction between carbon NPs and SARS-CoV-2 RNA fragments [197]
Amine-containing heterolipid NPs	116 physicochemical descriptors	PLS regression coupled with stepwise forward algorithm	Predict the pKa of the amine-containing heterolipid NPs [198]
Metal oxide NPs	Quantum-mechanical computations (such as molecular geometries), physicochemical descriptors (such as zeta-potential in water), and periodic table descriptors (such as electronegativity of each atom)	PLS regression, DecTrees, SVM, and logReg	Predict the inflammatory potential of metal oxide NPs [199]
Functionalized magneto-fluorescent NPs	Norm index descriptors (describing the structural characteristics of the involved NPs)	RF	Predict the cellular uptake of functionalized magneto-fluorescent NPs to PaCa2 cells. Provide guidance for the design and manufacture of safer nanomaterials [200]
Metal and metal oxide NPs	Structural information (such as core structure and material type), supported by physicochemical descriptors (such as zeta potential and average agglomerate size in media)	DecTrees, GBR, KNN, LinReg, RF, SVM, and XGBoost	Predict the cytotoxicity of metal and metal oxide NPs in zebrafish embryos [201]
Virtual carbon NP library	126 nanodescriptors (such as electronegativity of each atom)	KNN and RF	Predict cytotoxicity and inflammatory responses induced by PM2.5 [202]
Functionalized magneto-fluorescent NPs	Improved optimal quasi-SMILES-based descriptors	MC	Predict the cellular uptake of functionalized magneto-fluorescent NPs to PaCa2 and HUVEC cell lines [203]
Gold NPs	Optimal quasi-SMILES-based descriptors	MC	Predict the cellular uptake of gold NPs to A549 cells [204]
Functionalized magneto-fluorescent NPs	Optimal quasi-SMILES-based descriptors	MC	Develop self-consistent predictive models for the cellular uptake of functionalized magneto-fluorescent NPs to PaCa2 cells [205]
Metal oxide NPs	Optimal quasi-SMILES-based descriptors	MC	Predict the cell viability of different cell lines when exposed to metal oxide NPs [206]
ZnO NPs	Optimal quasi-SMILES-based descriptors	MC	Predict the toxicity of ZnO NPs in rats via intraperitoneal injections [207]
Metal oxide NPs	Optimal quasi-SMILES-based descriptors	MC	Predict the cell viability (expressed in %) and cytotoxicity (categorized as true or false) of different cell lines when exposed to 7 types of metal oxide NPs [208]
Cadmium QDs	Optimal quasi-SMILES-based descriptors	MC	Predict hepatic cell viability when exposed to cadmium QDs [209]
FDs	Structural information (such as polarizability), optimal quasi-SMILES-based descriptors, and physicochemical properties (obtained from Data Warrior)	MC and CPANN	Predict the binding score activity for 169 FDs related to 5 proteins classified as antidiabetes targets [210]
Metal-based nanomaterials	Optimal quasi-SMILES-based descriptors	MC	Predict the response of *Daphnia magna* when exposed to metal-based nanomaterials [211]

FDs: fullerene derivatives; QSAR: quantitative structure–activity relationships; NPs: nanoparticles; MLR: multiple linear regression; GNPs: gold nanoparticles; CNPs: carbon nanoparticles; PLS: partial least squares; SARS-CoV-2: severe acute respiratory syndrome coronavirus 2; SW-PLSR: PLS regression coupled with stepwise; SVM: support vector machine; RF: random forest; PaCa2: pancreatic cancer cells; KNN: k-nearest neighbors; MC: Monte Carlo; HUVEC: human umbilical vein endothelial cell; MO: metal oxide; ZnO: zinc oxide; CPANN: counter-propagation artificial neural network. ^1^ All models included follow the standardization and validation principles established by the Organisation for Economic Co-operation and Development (OECD) [189].

## Data Availability

Data are contained within the article or Appendix A.

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
