# Peer review of "Nanosafety: An Evolving Concept to Bring the Safest Possible Nanomaterials to Society and Environment"

_nanomaterials, 2022, doi:10.3390/nano12111810_

Round 1

Reviewer 1 Report

The submitted work meets all required criteria. I recommend accepting without changes. Only possible recommendations (at the discretion of the editors): too many citations and a large file (27 pages of work).

Author Response

We thank the reviewer for the positive evaluation of our work and for the suggestion. We believe all the references included in the manuscript are very relevant and, therefore, have decided to keep the manuscript in its original form

Reviewer 2 Report

Manuscript deals with the nanosafety issue as the concept aiming to bring the safest possible nanomaterials to society and environment. The authors presented a comprehensive review in which many essential toxicity aspects of nanomaterials were raised, including potential toxicity, immunotoxicity, and epigenotoxicity. The chapter concerning nanosensors, and organ-on-chip technologies is, in my opinion, especially interesting. LCA aspects related to the evaluation of nanomaterials constitute a perfect summary of discussed issues. To sum up, I have read the manuscript with great interest, and I recommend publishing it in Nanomaterials. The data combined in this review are of potential interest for pharmacy and materials sciences specialists. I would have only some minor suggestions for the Authors:

Page 17, line 623 – please consider “decrease”

Page 12, line 426 – please consider “is necessary”

Page 12, line 415 and line 429 - - please consider for 5mC “5-methylcytosine”

Page 11, line 365 – please consider “a lot of interest is focused on” or any relevant

Page 1, line 41 – please consider “already” within the expression “have been already …”

Page 2, Figure 1 – Don’t you think that a possible interaction should be presented between “aquatic organisms”, who ‘’bioaccumulate” NPs and humans? In my opinion, the relations “Bioaccumulation” – “Ecotoxicity” should be reconsidered. I would suggest considering “microorganisms” and “aquatic organisms” as blocks in this diagram and consider their impact on “humans”. “Effects on land animals” are also combined with “humans”. The food chain should be considered in this diagram otherwise, is says nothing.

Author Response

Manuscript deals with the nanosafety issue as the concept aiming to bring the safest possible nanomaterials to society and environment. The authors presented a comprehensive review in which many essential toxicity aspects of nanomaterials were raised, including potential toxicity, immunotoxicity, and epigenotoxicity. The chapter concerning nanosensors, and organ-on-chip technologies is, in my opinion, especially interesting. LCA aspects related to the evaluation of nanomaterials constitute a perfect summary of discussed issues. To sum up, I have read the manuscript with great interest, and I recommend publishing it in Nanomaterials. The data combined in this review are of potential interest for pharmacy and materials sciences specialists. I would have only some minor suggestions for the Authors:

Page 17, line 623 – please consider “decrease”

Page 12, line 426 – please consider “is necessary”

Page 12, line 415 and line 429 - - please consider for 5mC “5-methylcytosine”

Page 11, line 365 – please consider “a lot of interest is focused on” or any relevant

Page 1, line 41 – please consider “already” within the expression “have been already …”

We thank the reviewer for noticing these oversights. They have now been corrected.

Page 2, Figure 1 – Don’t you think that a possible interaction should be presented between “aquatic organisms”, who ‘’bioaccumulate” NPs and humans? In my opinion, the relations “Bioaccumulation” – “Ecotoxicity” should be reconsidered. I would suggest considering “microorganisms” and “aquatic organisms” as blocks in this diagram and consider their impact on “humans”. “Effects on land animals” are also combined with “humans”. The food chain should be considered in this diagram otherwise, is says nothing.

We have included a new version of Figure 1.

Dear reviewer, thank you very much for your comments. We have considered your suggestions and adopted most of them. Regarding figure 1, we present a new version that we think addresses your comment.

Reviewer 3 Report

The review is comprehensive, well organized and useful. I have the only recommendation about possible revision. While toxicological effects of ENM are well described, the review misses the comparison of various types of existing nanomaterilas for their toxic effect and discussion on the reasons of that. This can be added either as a new chapter or can be added as a discussion to a related part of the review.

Author Response

The review is comprehensive, well organized and useful. I have the only recommendation about possible revision. While toxicological effects of ENM are well described, the review misses the comparison of various types of existing nanomaterials for their toxic effect and discussion on the reasons of that. This can be added either as a new chapter or can be added as a discussion to a related part of the review.

Dear reviewer, thank you very much for your comment. We understand how important is to address the differences in toxicity of ENM, but the aim of this review is to provide an overview of the current status of the approaches used to evaluate ENM. Nonetheless, in our review we provide some insight on what you are suggesting. Specially at Table 4 we provide some information regarding specific nanomaterials and key biological outcomes evaluated. Also, in line with your comment, the information presented in Figure 7 and discussed following to that, the role of specific descriptors related to certain nanomaterials is addressed.
